# Live observation of the oviposition process in *Daphnia magna*

**Dohyong Lee, Ji Soo Nah, Jungbin Yoon, Won Kim\*, Kunsoo Rhee**  *

Department of Biological Sciences, Seoul National University, Seoul, Korea

\* rheek@snu.ac.kr (KR); wonkim@snu.ac.kr (WK)

## Abstract

In favorable conditions, *Daphnia magna* undergoes parthenogenesis to increase progeny production in a short time. However, in unfavorable conditions, *Daphnia* undergoes sexual reproduction instead and produces resting eggs. Here, we report live observations of the oviposition process in *Daphnia magna*. We observed that the cellular contents flowed irregularly through the narrow egg canal during oviposition. Amorphous ovarian eggs developed an oval shape immediately after oviposition and, eventually, a round shape. Oviposition of resting eggs occurred in a similar way. Based on the observations, we propose that, unlike *Drosophila* eggs, *Daphnia* eggs cannot maintain cytoplasmic integrity during oviposition. We also determined that the parthenogenetic eggs were activated within 20 min, as demonstrated by vitelline envelope formation. Therefore, it is plausible that the eggs of *Daphnia magna* may be activated by squeezing pressure during oviposition.

**Data Availability Statement:** All relevant data are within the manuscript and its Supporting Information files.

**Funding:** This work was supported by Marine Biotechnology Program (PJT200620) National Research Foundation of Korea (NRF-

## Introduction

Crustaceans of the genus *Daphnia* are used as model organisms in the fields of toxicology, ecology, and evolutionary biology [1]. *Daphnia* species exhibit an alternative reproductive mode. They usually undergo parthenogenesis to generate diploid female offspring without a fertilization event [2]. Parthenogenesis is maintained when surrounding conditions are favorable. However, when *Daphnia* are exposed to unfavorable conditions, male offspring are parthenogenetically produced, and sexual reproduction proceeds [2–4]. Various environmental factors can affect this process, such as temperature, photoperiod, food availability, and population density [2, 3, 5].

*Daphnia* produces different kinds of eggs in accord with its reproductive modes. In parthenogenesis, they mostly produce subitaneous eggs, or summer eggs, which can develop into organisms without the contribution of sperm [2, 3, 5]. However, in sexual reproduction, they produce resting eggs, or winter eggs, which differ greatly from subitaneous eggs, and an ephippium, which is a protective chamber for resting eggs [4, 5]. By laying resting eggs into the ephippium, they can remain in a dormant state and undergo activation when they encounter favorable conditions again. Subitaneous eggs generally outnumber resting eggs in a single oviposition event. *Daphnia magna* can produce up to eighty subitaneous eggs through parthenogenesis, while they produce only two resting eggs in an ephippium through sexual reproduction [2, 3, 5, 6].

2019R1A4A000000). The funders had no role in study design, data collection and analysis, decision to publish, or preparation of the manuscript.

**Competing interests:** The authors have declared that no competing interests exist.

Although the interval between oviposition events varies, depending on several factors, it usually takes about 3 to 4 days in favorable conditions [6]. Oocytes and nurse cells are indistinguishable in early stages, but only the oocytes accumulate yolk granules and oil droplets during maturation [7, 8]. It takes approximately 60 h for an oocyte to become a fully grown egg. When the eggs are ready to be ovulated, the animal undergoes the molting process, and oviposition follows 13 min after molting [2]. The eggs develop into neonates in the brood chamber and emerge from it before the subsequent molting process starts [1, 2].

In the present study, we recorded oviposition and egg activation in *Daphnia magna*. Live observations of oviposition revealed that ovarian components flowed through a very narrow egg canal. Egg activation also occurred after oviposition, as demonstrated by vitelline envelope (VE) formation.

## Materials and methods

### Animals

*Daphnia magna* was obtained from the National Institute of Environmental Research (Incheon, Korea). The animals were cultured in M4 medium and maintained under a 16 h:8 h light:dark photoperiod at 23ºC. The animals were fed with *Chlorella vulgaris*. Their density was controlled to 10–20 individuals in 700 ml to maintain only parthenogenetic females. To produce resting eggs, the animals were cultured in a high-density environment with up to 100 individuals in 700 ml of M4 medium.

### Live imaging with microscopes

We transferred *Daphnia magna* that underwent molting to a petri-dish with a small amount of M4 medium. Two different microscopes were used to record the oviposition process. A stereomicroscope (Leica S8AP0) was used to observe whole animals, while a light microscope (Olympus IX81) was used to focus on the egg canal. For recording, we used Las EZ 3.3.0 for the stereo-microscope and MetaMorph 7.6.5.0 for the light microscope.

### Histological analysis

For histological analysis, *Daphnia magna* was fixed with a 4% paraformaldehyde solution for 30 min. After serial sections were obtained from paraffin-embedded blocks, we performed H&E staining. The slides were observed with a light microscope (Olympus BX51). The connection between the ovarian egg and the extruded egg was easily disrupted when the individuals struggled in the fixative. To obtain intact sample, the animals were therefore pre-exposed to ice to minimize their writhing. To observe VE formation, we fixed *Daphnia magna* at the indicated time points after oviposition and determined the eggs with a VE.

## Results

### Oviposition of parthenogenetic and resting eggs

We observed the oviposition process in *Daphnia magna* with stereo- and light microscopes (Fig 1A, Fig 1B, S1 Movie and S2 Movie). The ovary near the oviposition site exhibited a round shape at the bottom, and the bottom-most egg started to protrude (Fig 1A and S1 Movie). During oviposition, the egg contents were extruded through the egg canal and accumulated outside the ovary (Fig 1A, Fig 1B, S1 Movie and S2 Movie). After completing oviposition, the eggs were transferred to the brood chamber surrounded by the carapace. It takes approximately 180 seconds for the completion of oviposition [2].

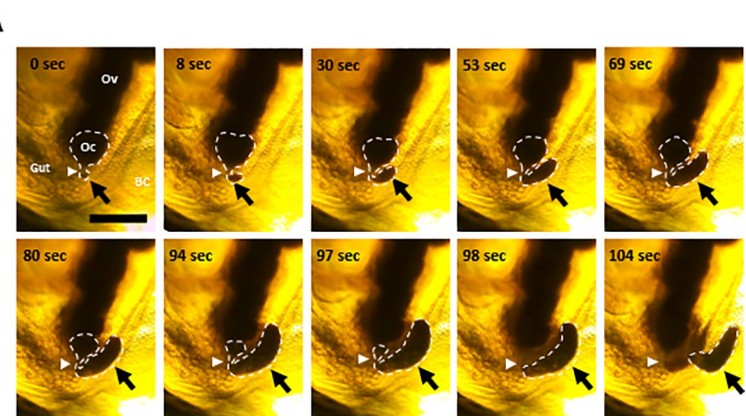

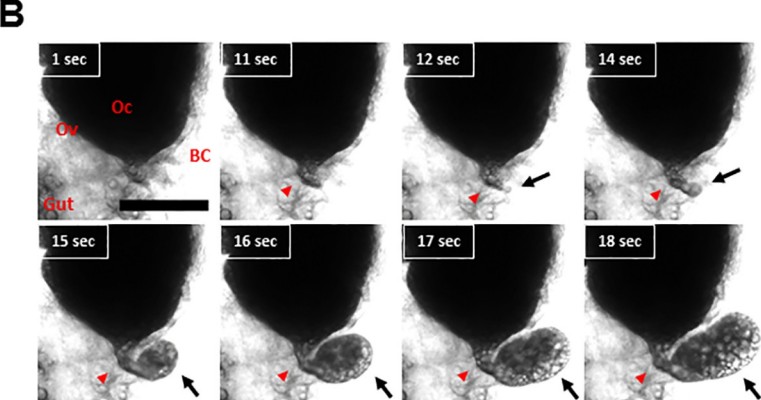

**Fig 1. Live images of parthenogenetic eggs undergoing oviposition in *Daphnia magna*.** The oviposition of parthenogenetic eggs was observed with a stereo- (A) and a light microscope (B). The dashed line indicates the outline of an ovulating egg. Arrowheads and arrows indicate the egg canal and extruded eggs, respectively. The time lapse after the initial observation is indicated in each panel. Scale bar, 200 μm. BC, Brood Chamber; Oc, Oocyte; Ov, Ovary. The movie files are deposited as S1 and S2 Movie.

We performed hematoxylin and eosin (H&E) staining to confirm the oviposition process of parthenogenetic eggs in *Daphnia magna*. As shown in Fig 2A and 2B, we were able to observe parthenogenetic eggs at the mid-point of oviposition. The diameter of the egg canal was approximately 24 μm (Fig 2A). Since the fully matured ovarian eggs had a diameter of 240–350 μm [1], they had to be extruded forcefully to pass through the egg canal.

During sexual reproduction, resting eggs are generated and placed in the ephippium. Ovaries under sexual reproduction are smaller than the case of parthenogenesis, and each ovary possesses one resting egg, total two eggs in ephippium [5]. We recorded the oviposition of resting eggs using a stereo- and a light microscope. The results showed that the contents of the resting eggs passed through a narrow egg canal and accumulated outside the ovary, as observed for parthenogenetic eggs (Fig 3A, Fig 3B, S3 Movie and S4 Movie).

### Formation of the vitelline envelope in parthenogenetic eggs

The VE formed as a result of egg activation [9, 10]. We performed H&E staining to observe VE formation in the parthenogenetic eggs of *Daphnia magna*. The VE was not visible in *Daphnia* eggs immediately after oviposition (Fig 4A①). Instead, it took time for the VE to start to

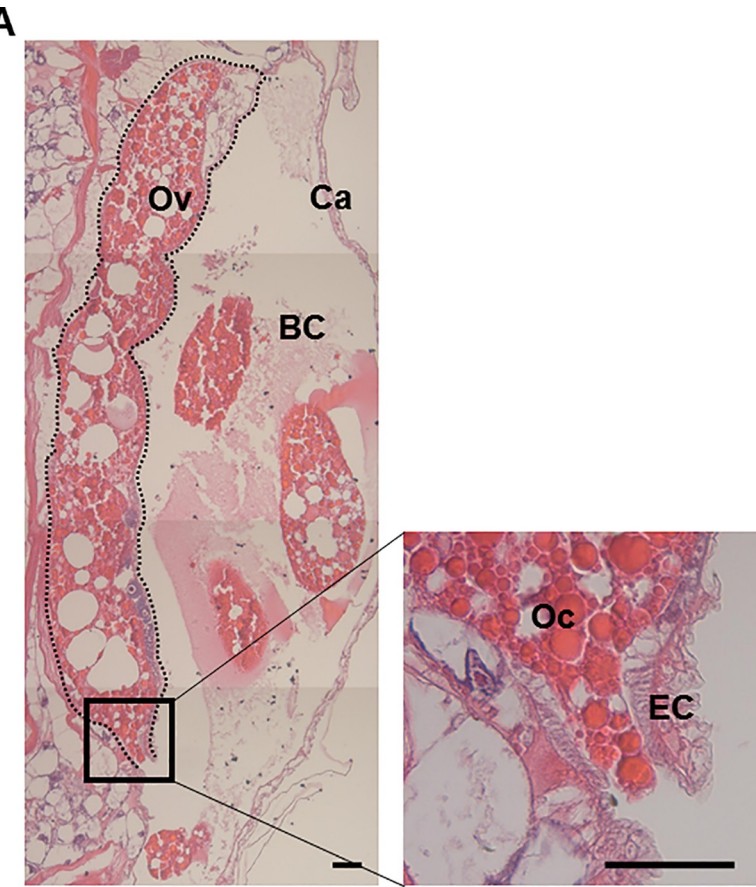

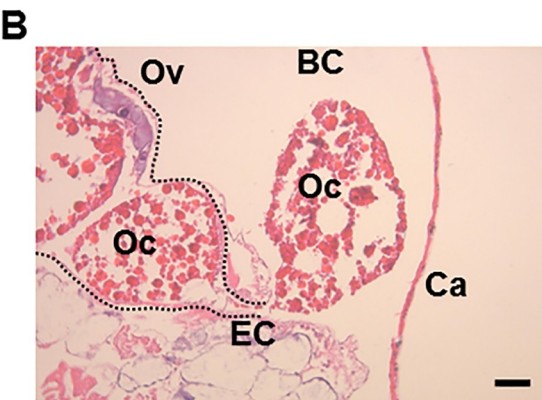

**Fig 2. H&E staining of parthenogenetic eggs undergoing oviposition in *Daphnia magna*.** (A) H&E staining of an oocyte undergoing oviposition. The boxed area in the left panel is magnified to show the enlarged egg canal. (B) H&E staining of an oocyte in which approximately a half of the contents were extruded. The dashed lined indicates the ovary. BC, Brood Chamber; Ov, Ovary; Ca, Carapace; Oc, Oocyte; EC, Egg Canal. Scale bar, 50 μm.

appear at the egg surface (Fig 4A②). The VE eventually surrounded the eggs (Fig 4A③) and was stabilized as a firm structure (Fig 4A④). To determine when the VE formed, we fixed the eggs at various time points after oviposition and carried out H&E staining. The results showed that VE formation started in approximately 20 min after the oviposition process and took approximately 20 min for completion (Fig 4B). Our results revealed that VE formation took some time in the parthenogenetic eggs of *Daphnia magna*.

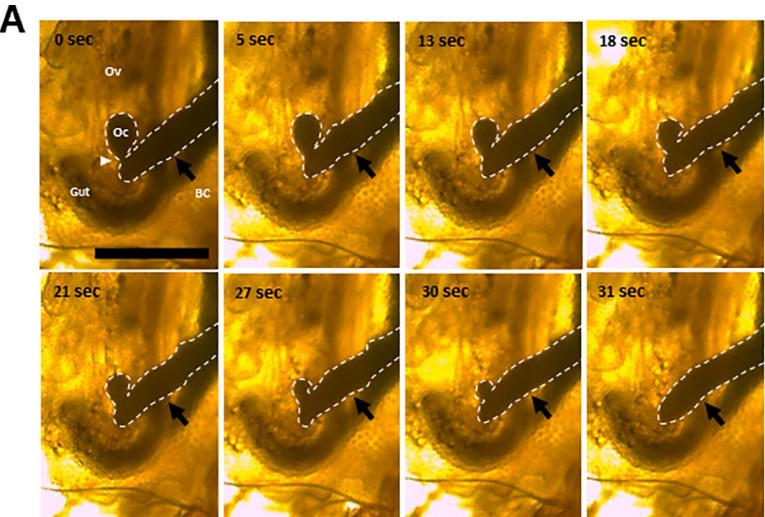

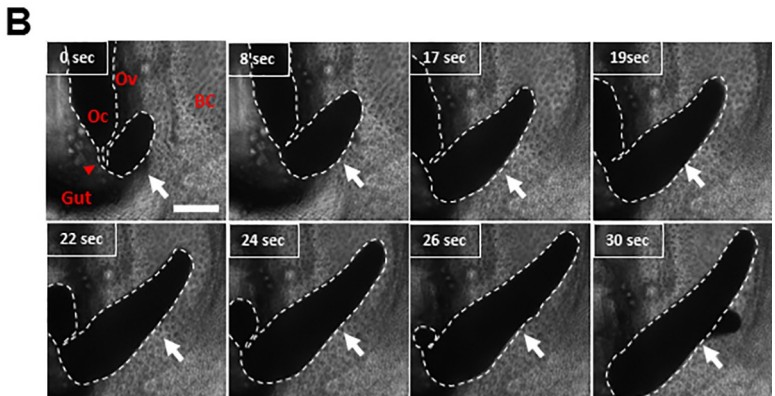

**Fig 3. Live images of resting eggs undergoing oviposition in *Daphnia magna*.** The oviposition of resting eggs was observed with a stereo- (A) and a light microscope (B). The dashed line indicates the outline of an ovulating egg. Arrowheads and arrows indicate the egg canal and extruded eggs, respectively. The time lapse after the initial observation is indicated in each panel. Scale bar, 200 μm. BC, Brood Chamber; Oc, Oocyte; Ov, Ovary. The movie files are deposited as **S3** and **S4 Movies**.

## Discussion

In this study, we used video microscopes to observe the oviposition process in *Daphnia magna*. During oviposition, *Daphnia* oocytes were forced through a narrow egg canal, and the egg contents flowed through the canal (Fig 5). After the completion of oviposition, the parthenogenetic eggs were activated after approximately 20 min, as evidenced by VE formation (Fig 5). We believe that this is the first report to provide live video recording of the oviposition process in *Daphnia magna*.

The body plan resides within the ovarian oocytes in *Drosophila melanogaster*. For example, mRNAs of *bicoid* and *nanos* are already located at the anterior and posterior ends, respectively, of pre-ovulatory oocytes [11]. *Drosophila* eggs maintain cellular integrity without significant disturbance when they pass through the narrow egg canal for ovulation [12]. However, our observations in *Daphnia magna* revealed that when the ovarian eggs passed through the

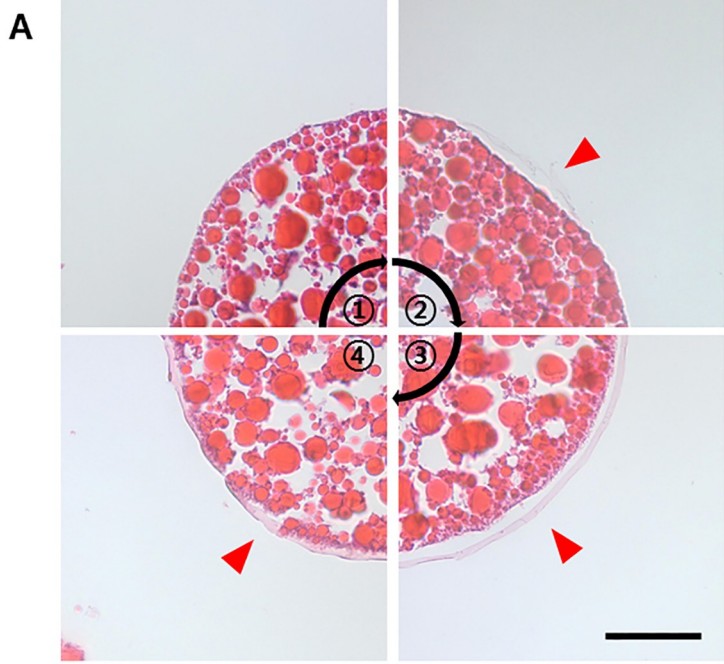

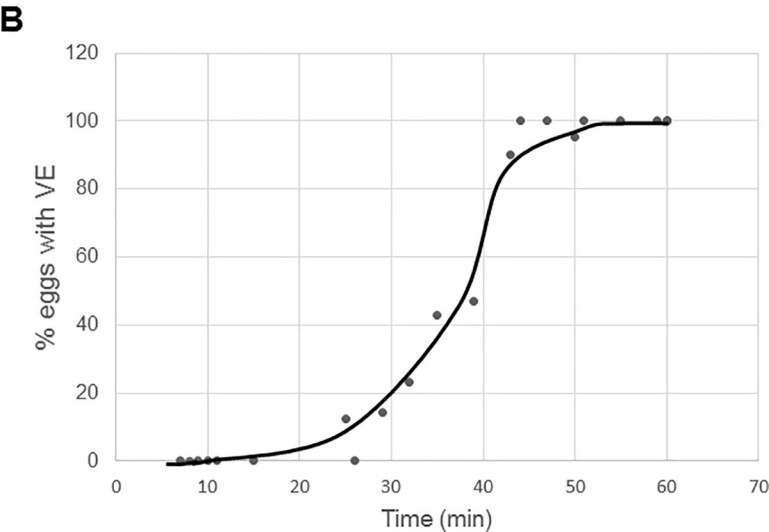

**Fig 4. Vitelline envelope formation after oviposition in *Daphnia magna*.** (A) H&E staining of parthenogenetic eggs forming a VE. ① Before VE formation; ② VE starting to form; ③ VE completely formed and inflated; ④ stabilized VE. Arrowheads indicate the VE. Scale bar, 50 μm. (B) The number of eggs with a VE was counted at the indicated time points after the initiation of oviposition.

narrow egg canal, their cellular contents flowed through the canal irregularly. The diameter of the egg canal is less than one-tenth of that of the pre-ovulatory eggs [1]. Amorphous ovarian eggs developed an oval shape immediately after oviposition and, eventually, a round shape (Fig 5). Therefore, we propose that, unlike *Drosophila* eggs, *Daphnia* eggs cannot maintain cytoplasmic integrity during oviposition. A few arthropods are known to have very narrow egg canals. For example, *Pimpla turionellae*, a haplodiploid wasp, has a narrow egg canal whose diameter is approximately one-third the width of the eggs [13]. The eggs of these wasps are extruded when they ovulate and undergo distortion to pass through the egg canal [14].

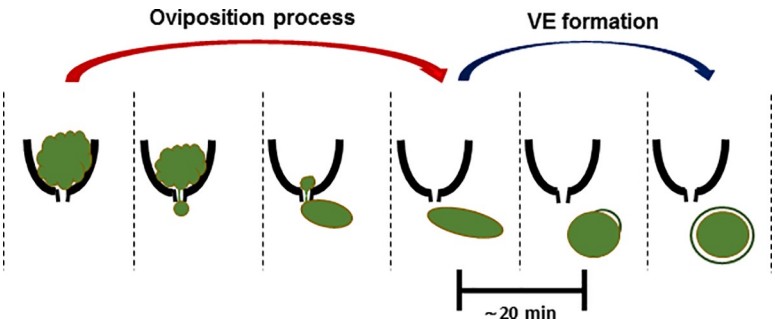

**Fig 5. Schematic illustration of the oviposition process and vitelline envelope formation in *Daphnia magna*.** An ovarian egg is extruded into the egg canal, and the egg contents accumulate outside the ovary. The egg undergoing extrusion continues to elongate until oviposition is completed. Thereafter, the VE forms within 20 min.

However, it remains to be investigated whether *Pimpla* oocytes maintain cellular integrity or not during oviposition.

In vertebrates and echinoderms, egg activation is caused by a fertilization event. However, in some case of arthropods, the fertilization event is not a precondition for their eggs to be activated [15]. As a result, arthropods can reproduce by using asexual methods such as parthenogenesis [16]. In the case of *Drosophila*, eggs are activated by pressure when they pass through the female reproductive tract [12, 17–19]. In the current study on *Daphnia magna*, we observed VE formation as a sign of egg activation during parthenogenesis. The VE of subitaneous egg was formed within approximately 20 min after oviposition, suggesting that activation of the parthenogenetic eggs occurred within this time period. The timing of VE formation has been reported in several arthropods, particularly in shrimps, in which the eggs are activated as soon as they are laid and exposed to seawater [16]. The eggs of *Penaeus monodon*, the black tiger shrimp, produce a complete hatching envelope within 15 min post-spawning [20, 21]. In addition, the eggs of *Sicyonia ingentis*, a prawn, start to form the hatching envelope approximately 25–30 min after spawning, and the process is complete in 40–45 min [22]. These reports indicate that it takes time for arthropod eggs to form the VE after the activation event. Therefore, it is likely that the parthenogenetic eggs of *Daphnia* are also activated prior to the VE formation. We speculate that the oviposition event in which Daphnid eggs pass through a narrow egg canal may be critical for egg activation. An increase in intracellular calcium levels triggers egg activation throughout the animal kingdom [16]. In the case of *Drosophila*, changes in intracellular calcium levels are induced by the action of the mechanosensitive ion channel during ovulation [15]. It remains to be determined when the change in intracellular calcium levels is induced in the parthenogenetic eggs of *Daphnia magna*.

## Supporting information

**S1 Movie. Live observation of parthenogenetic egg oviposition by stereo-microscope.** The oviposition process of a parthenogenetic egg in *Daphnia magna* was recorded using a stereo-microscope. The timing indicated in Fig 1A is matched with the timeline of this movie. (MP4)

**S2 Movie. Snapshots of parthenogenetic egg oviposition by light microscope.** The oviposition process of a parthenogenetic egg in *Daphnia magna* was recorded using a light microscope. The timing indicated in Fig 1B is matched with the timeline of this movie. (AVI)

**S3 Movie. Live observation of resting egg oviposition by stereo-microscope.** The oviposition process of a resting egg in *Daphnia magna* was recorded using a stereo-microscope. The timing indicated in Fig 3A is matched with the timeline of this movie.
(MP4)

**S4 Movie. Snapshots of resting egg oviposition by light microscope.** The oviposition process of a resting egg in *Daphnia magna* was recorded using a light microscope. The timing indicated in Fig 3B is matched with the timeline of this movie.
(AVI)

## Author Contributions

**Conceptualization:** Dohyong Lee, Jungbin Yoon, Won Kim, Kunsoo Rhee.

**Funding acquisition:** Won Kim, Kunsoo Rhee.

**Investigation:** Dohyong Lee, Ji Soo Nah, Kunsoo Rhee.

**Methodology:** Jungbin Yoon.

**Project administration:** Won Kim, Kunsoo Rhee.

**Resources:** Ji Soo Nah.

**Supervision:** Jungbin Yoon, Won Kim, Kunsoo Rhee.

**Validation:** Kunsoo Rhee.

**Writing – original draft:** Dohyong Lee, Kunsoo Rhee.

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
