## [Decision Letter · Decision Letter 0]

21 Aug 2019

PONE-D-19-15062

Live observation of the oviposition process in Daphnia magna

PLOS ONE

Dear Professor Rhee,

Thank you for submitting your manuscript to PLOS ONE. After careful consideration, we feel that it has merit but does not fully meet PLOS ONE’s publication criteria as it currently stands. Therefore, we invite you to submit a revised version of the manuscript that addresses the points raised during the review process.

We would appreciate receiving your revised manuscript by Oct 05 2019 11:59PM. To enhance the reproducibility of your results, we recommend that if applicable you deposit your laboratory protocols in protocols.io, where a protocol can be assigned its own identifier (DOI) such that it can be cited independently in the future. For instructions see: http://journals.plos.org/plosone/s/submission-guidelines#loc-laboratory-protocols

We look forward to receiving your revised manuscript.

Kind regards,

Gao-Feng Qiu

Academic Editor

PLOS ONE

Journal Requirements:

1. Thank you for including the following funding information within your manuscript; "This research was supported by the Marine Biotechnology Program (PJT200620) and

Basic Science Research Program through the National Research Foundation of Korea (NRF-

2019R1A4A000000)."

We note that you have provided funding information that is not currently declared in your Funding Statement. However, funding information should not appear within your manuscript. We will only publish funding information present in the Funding Statement section of the online submission form.

Reviewers' comments:

Reviewer's Responses to Questions

**Comments to the Author**

1. Is the manuscript technically sound, and do the data support the conclusions?

Reviewer #1: Yes

Reviewer #2: Yes

2. Has the statistical analysis been performed appropriately and rigorously? 

Reviewer #1: N/A

Reviewer #2: N/A

3. Have the authors made all data underlying the findings in their manuscript fully available?

Reviewer #1: Yes

Reviewer #2: Yes

4. Is the manuscript presented in an intelligible fashion and written in standard English?

Reviewer #1: Yes

Reviewer #2: Yes

5. Review Comments to the Author

Reviewer #1: While the scope is narrow, this descriptive morphological study revealed some interesting details concerning the process of egg deposition in daphnia. The following are detailed comments.

1. Some statements regarding basic daphnia biology are not accurate. Second paragraph of introduction, during parthenogenic reproduction, Daphnia magna can produce up to ~80 neonates depending on availability of food, ambient temperatures, age, etc. The range of 20-30 is too narrow. The authors are suggested to check with the classic review by Hebert (1978). Third paragraph of introduction, the brood interval is not fixed at 3 days but varies depending on several factors.

2. Third paragraph of results, each ephippium is known to contain two resting eggs. Why one resting egg?

3. The first sentence of the 4th paragraph needs to be backed up by references. So, according to this statement egg activation occurs before vitelline envelope formation? Also, what did the authors exactly mean by egg activation?

4. Second last sentence of the 4th paragraph in the results section, “to form after approximately 20 min” is not clear. It meant to be “approximately 20 min after egg deposition”?

5. Second sentence of the third paragraph in discussion, saying “arthropod eggs” is too general. Consider “the eggs of some arthropods”. Same paragraph, “The VE formed within…….”.

6. There does not appear to be good logic in the last 4 sentences of the 3rd paragraph in discussion. There is a lack of causal relationship between the three sentences concerning shrimp and prawn and the last sentence. What does a delay in VE formation observed in a shrimp and a prawn have anything to do with the speculation that egg activation in daphnia is triggered by squeezing in the oviduct?

7. References are not standardized. Each item should be in the same format and the Latin names italicized.

Reviewer #2: This article studied the oviposition process and egg activation of Daphnia magna from the perspective of morphology. This study is not suitable for submission to this journal and is recommended for submission to professional morphology journals.

6. PLOS authors have the option to publish the peer review history of their article (what does this mean?). If published, this will include your full peer review and any attached files.

Reviewer #1: No

Reviewer #2: No

---

## [Author Response · Author response to Decision Letter 0]

4 Oct 2019

Reviewer #1: While the scope is narrow, this descriptive morphological study revealed some interesting details concerning the process of egg deposition in daphnia. The following are detailed comments.

1 Some statements regarding basic daphnia biology are not accurate. Second paragraph of introduction, during parthenogenic reproduction, Daphnia magna can produce up to ~80 neonates depending on availability of food, ambient temperatures, age, etc. The range of 20-30 is too narrow. The authors are suggested to check with the classic review by Hebert (1978). Third paragraph of introduction, the brood interval is not fixed at 3 days but varies depending on several factors.

We are grateful of helpful comments of the reviewer. We carefully took a look at the review by Hebert (1978) and corrected as suggested. Ebert (2005) also indicated that an adult female may produce a clutch of eggs every 3 to 4 days until her death. 

2. Third paragraph of results, each ephippium is known to contain two resting eggs. Why one resting egg?

We understand that each ovary produce one resting egg. Since an ephippium originate from two ovaries, it contains two resting eggs. The text was corrected as suggested. 

3. The first sentence of the 4th paragraph needs to be backed up by references. So, according to this statement egg activation occurs before vitelline envelope formation? Also, what did the authors exactly mean by egg activation?

As suggested, we backed up the sentence with two references (Anderson, 1967; Masuda et al., 1991). The most critical event of egg activation may be the elevation of intracellular calcium levels (Horner and Wolfner, Dev Dyn 237:527-544, 2008). Egg activation in Drosophila is known to occur during ovulation (Kaneuchi et al., PNAS 112:791-796, 2015). Egg activation of shrimp was also reported during spawning and formation of vitelline envelope follows after egg activation (Pongtippatee-Taweepreda et al, 2004; Pongtippatee et al., 2012). VE formation is followed after egg activation (Anderson, 1967; Masuda et al., 1991).

4. Second last sentence of the 4th paragraph in the results section, “to form after approximately 20 min” is not clear. It meant to be “approximately 20 min after egg deposition”?

I appreciate for your correction. We changed the sentence as suggested.

5. Second sentence of the third paragraph in discussion, saying “arthropod eggs” is too general. Consider “the eggs of some arthropods”. Same paragraph, “The VE formed within…….”.

I appreciate for your careful reading. We changed the sentence as suggested.

6. There does not appear to be good logic in the last 4 sentences of the 3rd paragraph in discussion. There is a lack of causal relationship between the three sentences concerning shrimp and prawn and the last sentence. What does a delay in VE formation observed in a shrimp and a prawn have anything to do with the speculation that egg activation in daphnia is triggered by squeezing in the oviduct?

It is likely that egg activation in Daphnia also precedes VE formation. We speculate that Daphnid eggs are activated during oviposition which is the most visible event prior to VE formation. In fact, Drosophila eggs are activated by pressure when they pass through the reproductive tract (Horner and Wolfner, 2008). We carefully rewrote the sentences to emphasize our points. 

7. References are not standardized. Each item should be in the same format and the Latin names italicized.

We formatted the whole manuscript, following strictly the instruction of Plos One. We also standardized the list of references as suggested. 

 

Reviewer #2: This article studied the oviposition process and egg activation of Daphnia magna from the perspective of morphology. This study is not suitable for submission to this journal and is recommended for submission to professional morphology journals.

We admit that this manuscript is descriptive. However, our works provide an insight how a parthenogenic egg starts development after oviposition. It is known that developmental patterns of embryos in many species are already fixed in oocytes. For example, the anterior-posterior and dorsal-ventral axes of Drosophila embryos are determined during oogenesis. Even if Drosophila eggs pass through a tight reproductive track for ovulation, their cytoplasmic arrangement should not be disturbed. However, the oviposition pattern of the parthenogenic Daphnia egg does not allow us to imagine that the developmental axis are maintained during oviposition. What we observed is that the egg contents flow through a narrow egg canal. After oviposition, Daphnia eggs are known to follow a strict developmental pattern. Significance of this work may be that we raise a possibility that developmental axis of Daphnia may not be fixed during oogenesis but be established after oviposition. Since our observations provide an insight on a new way for developmental pattern formation, we believe that this manuscript brings a general interest in the field of animal development as well as Daphnia research.

---

## [Decision Letter · Decision Letter 1]

14 Oct 2019

Live observation of the oviposition process in Daphnia magna

PONE-D-19-15062R1

Dear Dr. Rhee,

We are pleased to inform you that your manuscript has been judged scientifically suitable for publication and will be formally accepted for publication once it complies with all outstanding technical requirements.

With kind regards,

Gao-Feng Qiu

Academic Editor

PLOS ONE

Additional Editor Comments (optional):

Reviewers' comments:

Reviewer's Responses to Questions

**Comments to the Author**

1. If the authors have adequately addressed your comments raised in a previous round of review and you feel that this manuscript is now acceptable for publication, you may indicate that here to bypass the “Comments to the Author” section, enter your conflict of interest statement in the “Confidential to Editor” section, and submit your "Accept" recommendation.

Reviewer #1: All comments have been addressed

2. Is the manuscript technically sound, and do the data support the conclusions?

Reviewer #1: Yes

3. Has the statistical analysis been performed appropriately and rigorously? 

Reviewer #1: N/A

4. Have the authors made all data underlying the findings in their manuscript fully available?

Reviewer #1: Yes

5. Is the manuscript presented in an intelligible fashion and written in standard English?

Reviewer #1: Yes

6. Review Comments to the Author

Reviewer #1: The authors responded to reviewers' comments and concerns pretty well.

Couple of typos:

1. Page line 9 from top, "smaller than the case of" to "smaller than those for".

2. Page line 4 from bottom, "Daphnid" to "daphnid".

7. PLOS authors have the option to publish the peer review history of their article (what does this mean?). If published, this will include your full peer review and any attached files.

Reviewer #1: No

---

## [Editor Report · Acceptance letter]

22 Oct 2019

PONE-D-19-15062R1 

Live observation of the oviposition process in *Daphnia magna*

Dear Dr. Rhee:

I am pleased to inform you that your manuscript has been deemed suitable for publication in PLOS ONE. Congratulations! Your manuscript is now with our production department. 

With kind regards,

on behalf of

Prof. Gao-Feng Qiu 

Academic Editor

PLOS ONE